# Amino Acid Polymorphisms in the Basic Region of Meq of Vaccine Strain CVI988 Drastically Diminish the Virulence of Marek’s Disease Virus

**DOI:** 10.3390/v17070907

**Published:** 2025-06-26

**Authors:** Jumpei Sato, Yoshinosuke Motai, Shunsuke Yamagami, Aoi Kurokawa, Shwe Yee Win, Fumiya Horio, Hikaru Saeki, Naoya Maekawa, Tomohiro Okagawa, Satoru Konnai, Kazuhiko Ohashi, Shiro Murata

**Affiliations:** 1Department of Disease Control, Faculty of Veterinary Medicine, Hokkaido University, Sapporo 060-0818, Japan; 2National Institute of Animal Health, National Agriculture and Food Research Organization, Tsukuba 062-0045, Japan; 3Department of Advanced Pharmaceutics, Faculty of Veterinary Medicine, Hokkaido University, Sapporo 060-0818, Japan; 4Institute for Vaccine Research and Development, Hokkaido University, Sapporo 060-0818, Japan; 5Veterinary Research Unit, International Institute for Zoonosis Control, Hokkaido University, Sapporo 001-0020, Japan; 6International Affairs Office, Faculty of Veterinary Medicine, Hokkaido University, Sapporo 060-0818, Japan

**Keywords:** Marek’s disease virus, Meq, CVI988

## Abstract

Marek’s disease virus (MDV) is the etiological agent of Marek’s disease (MD), a lymphoproliferative disorder in chickens. Polymorphisms in the MDV-encoded oncoprotein Meq are shared among field strains and correlate with their virulence. The attenuated vaccine strain CVI988 harbors unique amino acid polymorphisms in Meq, particularly at positions 71, 77, and 326. In this study, we investigated the impact of these polymorphisms on Meq protein function and MDV virulence. Reporter assays revealed that the substitutions, particularly A71S and K77E, markedly impaired the transcriptional regulatory activity of Meq. To evaluate their effect on virulence, we generated a recombinant MDV based on the very virulent RB-1B strain, encoding Meq with A71S and K77E substitutions (rRB-1B_Meq71/77). Chickens infected with rRB-1B_Meq71/77 developed neither clinical signs nor lymphomas. Flow cytometry revealed no expansion of infected cells in this group, but a marked increase in CD8^+^ T and γδ T cells during early infection. Histopathological analysis also confirmed the absence of MD-associated lesions. These findings demonstrate that the polymorphisms at positions 71 and 77 in the CVI988 strain are sufficient to abolish MDV virulence. This study provides insight into the molecular basis of MDV virulence and informs the strategy for the design of more effective vaccines.

## 1. Introduction

Marek’s disease virus (MDV) is the etiological agent of Marek’s disease (MD), characterized by the formation of malignant lymphomas and neurological disorders. MD has caused severe economic losses to the poultry industry; however, its occurrence has been reduced through the vaccination of chickens with attenuated MDV and/or non-pathogenic vaccine viruses [1,2]. Nevertheless, the virulence of field strains has continued to increase, resulting in sporadic MD outbreaks even in vaccinated flocks in certain regions [3,4,5,6,7,8]. MDV virulence is currently categorized into four pathotypes based on the ability to induce clinical signs in vaccinated chickens: mild, virulent, very virulent (vvMDV), and very virulent plus (vv+MDV) [9]. Field strains recently isolated from China and Australia have been reported to exhibit high virulence equivalent to that of vvMDV or vv+MDV strains [10,11,12]. Furthermore, genetic analyses have revealed that vvMDV and vv+MDV strains continue to circulate widely across continents, including North America, Europe, and Africa [11,12,13,14]. Consequently, there is ongoing concern regarding potential future outbreaks caused by highly virulent MDV strains.

Although MDV is classified as an alphaherpesvirus (subfamily *Alphaherpesvirinae*, genus *Mardivirus*, species *Gallid alphaherpesvirus 2*), its biological characteristics more closely resemble those of lymphotropic oncogenic gammaherpesviruses, such as Epstein-Barr virus and Kaposi’s sarcoma-associated herpesvirus [15]. Following entry into the respiratory tract, MDV infects various immune cells, including macrophages, dendritic cells, and B cells, which are thought to facilitate viral transport to primary lymphoid organs [16]. Within 48 h post-infection, MDV can be detected in the spleen, thymus, and bursa of Fabricius [16,17], where it infects B cells, natural killer (NK) cells, and T cells [18,19,20]. In particular, MDV induces transient atrophy of the thymus and bursa of Fabricius by suppressing T and B cell proliferation and promoting apoptosis [21,22]. After lytic replication, MDV establishes latent infection in CD4^+^ T cells, with a subset of these cells subsequently undergoing transformation [23]. This transformation results in neoplastic T cell lesions in visceral organs, whereas infiltrating lymphocytes may induce peripheral nerve edema, causing paralysis [24,25]. MDV-transformed tumor cells primarily consist of CD4^+^ T cells, arising from (oligo)clonal expansion of transformed cells [23,26,27]. Consequently, an increased proportion of transformed CD4^+^ T cells is observed in the peripheral blood during the late phase of MDV infection [28]. Infected lymphocytes also transport MDV to the skin, where the virus replicates within the feather follicle epithelium and is subsequently shed into the environment, contributing to viral transmission [29].

Throughout the infection cycle, MDV elicits innate and adaptive immune responses involving multiple immune cell types, including CD8^+^ T cells, γδ T cells, macrophages, and NK cells [30,31,32,33,34]. Among these, CD8^+^ T cells serve as key effectors of cellular immunity, targeting infected and transformed cells. CD8^+^ T cells can recognize MDV-derived antigens and exert cytotoxic activity [35]. Notably, CD8αα^+^ T cells increase during the proliferative phase of MDV infection [36] and expand rapidly following booster immunization with CVI988 [34], suggesting the presence of MDV-specific memory T cells within this subset. In contrast, γδ T cells comprise a substantial fraction of the T cell compartment in chickens, and their protective role against MDV infection has been demonstrated in γδ T cell-deficient chickens [20]. Their frequency increases during the early lytic phase of MDV infection [31]. Furthermore, the proportion of CD8α^+^ γδ T cells expressing IFN-γ is elevated during early cytolytic infection in chickens superinfected with RB-1B after vaccination [31], highlighting their pivotal role in initiating a cytokine-mediated antiviral response during innate immunity.

The MDV oncogene *meq* encodes the 339-amino-acid Meq protein, which is expressed during the lytic and latent phases of infection [37]. A *meq*-deleted recombinant MDV (rMDV) failed to induce lymphoma in infected chickens, indicating that Meq plays a critical role in MDV-induced transformation [38]. Meq is a basic leucine zipper (bZIP) transcription factor that regulates the expression of various viral and host genes associated with pathogenesis. It contains a basic region and ZIP motif in the N-terminal domain and a transactivation domain characterized by proline-rich repeats (PRRs) in the C-terminus [37]. The basic region mediates nuclear and nucleolar localization and DNA binding [37]. The bZIP motif resembles those found in the oncoproteins Jun and Fos, allowing Meq to form homodimers or heterodimers with AP-1 family proteins, such as c-Jun, JunB, and Fos [39]. Meq upregulates genes in the *v-Jun* transformation pathway, including *JTAP-1*, *JAC*, and *HB-EGF*, in transformed DF-1 cells [40]. Furthermore, Meq interacts with several host proteins, including p53 and STING, inhibiting their functions [41,42]. Thus, Meq exerts diverse functions by interacting with proteins involved in tumorigenesis and immune evasion.

Certain amino acid residues in the basic region and PRRs of the transactivation domain are conserved among field strains [43]. For instance, highly virulent U.S. strains such as RB-1B and Md-5 contain alanine at position 71, lysine at position 77, and threonine at position 326 (Table 1). Conversely, the CVI988 strain, an attenuated vaccine derived from a low-virulence MDV strain, possesses serine at position 71, glutamic acid at position 77, and isoleucine at position 385/386. This corresponds to position 326 of the 339-amino acid Meq protein as a result of the insertion of 59/60 additional amino acids in the transactivation domain; this variant is referred to as the long isoform of Meq (L-Meq) [43]. In experimental infections, rMDV expressing Meq from highly virulent strains exhibited significantly higher mortality than rMDV expressing Meq from less virulent strains [44].

The CVI988 strain, developed by attenuation of a low-virulence MDV strain isolated from a chicken with antibodies against MDV but no clinical signs, is globally used as the current gold standard vaccine strain [45]. CVI988 provides effective protection against virulent MDV strains by inducing long-lasting immunity and antibody production [45]. CVI988 consists of multiple subpopulations, some encoding the 339-amino-acid Meq (CVI988-Meq) and others encoding a 399-amino-acid Meq variant (CVI988-L-Meq) [46]. Initially, the insertion sequence was thought to contribute to low virulence. However, previous findings demonstrated that this insertion sequence enhances virulence in experimental infections using RB-1B-based rMDVs [47]. Moreover, an RB-1B-based rMDV expressing wild-type CVI988-L-Meq exhibited virulence comparable to the wild-type RB-1B strain [48]. Additionally, an rMDV expressing a chimeric Meq composed of the N-terminal region of CVI988-Meq and the C-terminal region of Md-5-Meq exhibited significantly reduced virulence compared to the wild-type Md-5 strain [49]. These findings suggest that amino acid polymorphisms in the Meq basic region, along with other viral factors, contribute to the low virulence of CVI988. However, the precise impact of these polymorphisms on virulence and the mechanisms underlying CVI988 low-virulence remain unclear.

This study investigated the effects of polymorphisms observed in the basic region of CVI988-Meq on virulence and transcriptional function. To assess their impact on transcriptional regulation, we analyzed the transcriptional regulatory activity of Meq variants using reporter assays. Additionally, we compared the virulence of rMDVs harboring RB-1B-Meq and Meq71/77 (containing serine and glutamic acid at positions 71 and 77, respectively) in experimental infections. Furthermore, we examined the dynamics of MDV-infected cells in parallel with the kinetics of CD8^+^ T and γδ T cells in rMDV-infected chickens.

## 2. Materials and Methods

### 2.1. Ethics Statement

All animal experiments were approved by the Institutional Animal Care and Use Committee of Hokkaido University (approval number 22-0088). All procedures were conducted in accordance with the relevant guidelines and regulations of the Faculty of Veterinary Medicine, Hokkaido University, which is fully accredited by the Association for Assessment and Accreditation of Laboratory Animal Care International.

### 2.2. Plasmids

Expression plasmids for each Meq isoform with mutations were constructed and the mutations were introduced via site-directed mutagenesis, as previously described [47]. The open reading frame of *meq* derived from RB-1B (accession number: HM488349.1) was amplified and cloned into the pCI-neo vector (Promega, Madison, WI, USA). Alanine-to-serine, lysine-to-glutamic acid, and threonine-to-isoleucine substitutions were introduced at positions 71, 77, and 326 of Meq, respectively, using the primers listed in Table 2. The resultant proteins were designated Meq(A71S), Meq(K77E), and Meq(T326I). A triple mutant with all three substitutions was also constructed and designated as Meq(Triple). For the transactivation activity assay, we constructed a c-Jun expression plasmid (38) and reporter plasmids by inserting the promoter regions of pp38, pp14, icp4, gb, chicken cd30, chicken bcl-2, and chicken il-2 upstream of the firefly luciferase-coding region in the pGL3-Basic vector (Promega, Madison, WI, USA), using the primers listed in Table 3 (38). The pRL-TK *Renilla* luciferase plasmid (Promega, Madison, WI, USA) was used as an internal control.

### 2.3. Dual-Luciferase Reporter Assay

DF-1 cells were seeded in 24-well plates at a density of 2.0 × 10^5^ cells/well and incubated for 24 h. For reporter assays using icp4, gb, chicken cd30, chicken bcl-2, and chicken il-2 promoters, each well was transfected with 300 ng of the Meq expression plasmid, 200 ng of the c-Jun expression plasmid, 500 ng of the reporter plasmid, and 5 ng of pRL-TK using Lipofectamine 2000 (Thermo Fisher Scientific, Waltham, MA, USA), according to the manufacturer’s instructions. For the reporter assay using pp38 and pp14 promoters, 300 ng of the expression plasmid, 500 ng of the reporter plasmid, and 5 ng of pRL-TK were transfected per well. Cells were lysed 24 h post-transfection using 1× Passive Lysis Buffer (Promega, Madison, WI, USA), and luciferase activity was measured using the Dual-Luciferase Reporter Assay System (Promega, Madison, WI, USA) using a Luminescencer AB-2350 Phelios (Atto Corp., Tokyo, Japan). Firefly luciferase activity was normalized to *Renilla* luciferase activity, and the results were reported relative to the luciferase activity of cells transfected with the pCI-neo vector without the c-Jun expression vector. Three independent experiments were conducted in triplicate.

### 2.4. Cells

Chicken embryo fibroblasts (CEFs) were obtained from 10-day-old fertilized eggs (Iwamura Hatchery Co., Ltd., Niigata, Japan), as previously described [55]. CEFs were cultured in Eagle’s minimal essential medium (Nissui Pharmaceutical Co., Ltd., Tokyo, Japan) supplemented with 10% tryptose phosphate broth (Difco Laboratories, Detroit, USA), 0.03% L-glutamine, 100 U/mL penicillin, 100 μg/mL streptomycin, 10% calf serum (Sigma-Aldrich, St. Louis, MO, USA), and 0.1% NaHCO_3_. DF-1 cells, an immortalized chicken fibroblast cell line, were maintained in Dulbecco’s modified Eagle’s medium (FUJIFILM Wako Pure Chemical Corporation, Osaka, Japan) containing 0.03% L-glutamine, 100 U/mL penicillin, 100 μg/mL streptomycin, and 10% fetal bovine serum (MP biomedicals, Santa Ana, CA, USA), and incubated at 39 °C in a humidified atmosphere with 5% CO_2_.

### 2.5. Generation of Recombinant Viruses

To investigate the effect of polymorphisms at positions 71 and 77 on virulence, alanine-to-serine and lysine-to-glutamic acid substitutions were introduced into RB-1B-*meq* (designated *meq*71/77) cloned in the pCI-neo vector via site-directed mutagenesis using the primers listed in Table 2. An RB-1B-based rMDV encoding Meq71/77 was then generated as previously described [44,47,48,56]. To construct rMDVs expressing Meq or Meq71/77, a bacterial artificial chromosome (BAC) plasmid containing the RB-1B genome with a deletion in most of the internal repeat long (IRL) region (pRB-1B_ΔIRL) was used [57]. The native *meq* gene in the terminal repeat long (TRL) was partially deleted (pRB-1B_ΔIRL_Δmeq), and either RB-1B-*meq* or *meq*71/77 was inserted into the TRL *meq* locus via two-step Red-mediated mutagenesis [58,59]. Recombinant BAC plasmids were screened using restriction fragment length polymorphism (RFLP) analysis with BamHI-HF (New England Biolabs Japan Inc., Tokyo, Japan) and HindIII-HF (New England Biolabs Japan Inc.), followed by electrophoresis on 0.8% agarose gel (Appendix A). The insertion of each *meq* isoform was confirmed by PCR and DNA sequencing. The resulting BAC plasmids were designated as pRB-1B_ΔIRL_Meq and pRB-1B_ΔIRL_Meq71/77, respectively. The BAC plasmids were transfected into CEFs using the CalPhos Mammalian Transfection Kit (Takara Bio Inc., Kyoto, Japan). The pCAGGS-Cre plasmid (Gene Bridges GmbH, Heidelberg, Germany) was co-transfected to remove the BAC cassette from the viral genome. The reconstituted recombinant viruses were designated rRB-1B_Meq and rRB-1B_Meq71/77. Viruses were propagated in CEFs and stored at –80 °C in Cell Banker 1 (Nippon Zenyaku Kogyo Co., Ltd., Fukushima, Japan). Since the IRL region is rapidly restored based on the sequence in the TRL region containing the RB-1B-*meq* or *meq*71/77 during viral reconstitution [60,61], restoration of the IRL and deletion of the BAC sequence were confirmed by PCR as previously reported [61]. The obtained recombinant viruses were named as rRB-1B_Meq and rRB-1B_Meq71/77, respectively. Each virus was titrated using plaque assays as previously described [57,62].

### 2.6. In Vitro Replication of the Recombinant Viruses

Confluent monolayers of CEFs seeded in 12-well plates were infected with 50 plaque-forming units (pfu) of recombinant virus. Infected cells were collected daily for 5 days. Total cellular DNA was extracted using the DNeasy Blood and Tissue Kit (Qiagen, Hilden, Germany). Viral load was quantified by quantitative PCR (qPCR) to assess replication kinetics. Primer sets used in the qPCR analysis are listed in Table 4. Viral loads were calculated as the mean of three independent cultures, with all experiments performed in triplicate.

### 2.7. Expression of Meq mRNA in Cells Infected with Recombinant Viruses in Vitro

CEFs were seeded in 12-well plates and infected with 50 pfu of recombinant viruses. Infected cells were collected daily for 5 days. Total RNA was extracted using TRI Reagent (Molecular Research Center, Inc., Cincinnati, OH, USA) according to the manufacturer’s instructions. The RNA was treated with DNase I (Promega, Madison, WI, USA) to remove residual genomic DNA, and cDNA was synthesized using PrimeScript Reverse Transcriptase (Takara Bio Inc., Kyoto, Japan). The expression levels of each *meq* isoform were quantified by qPCR using TB Green Premix DimerEraser (Takara Bio Inc., Kyoto, Japan). The expression of *β-actin* was also quantified by qPCR and used as an internal control. *meq* mRNA expression was presented as the ratio of *meq* mRNA concentration to that of *β-actin* mRNA. The primer sets used for qPCR are listed in Table 4.

### 2.8. In Vivo Characterization of Recombinant Viruses

For experimental infection, fertilized eggs from commercial White Leghorn chickens (Iwamura Hatchery Co., Ltd., Niigata, Japan) were hatched and raised in isolators at the animal facility of the Faculty of Veterinary Medicine, Hokkaido University. The following animal experiments were conducted to characterize the recombinant viruses. Attending veterinarians monitored the health status of chickens daily throughout the experimental period. This experimental infection was conducted by research staff who had received comprehensive training through an animal experimentation program administered by Hokkaido University. We set the humane endpoint at the onset of neurological symptoms, such as leg paralysis and depression, and chickens reaching this stage were euthanized on the same day. After deep general anesthesia with isoflurane (Zoetis Japan, Tokyo, Japan), all chickens were euthanized by cardiac blood collection, and heparinized blood was collected.

#### 2.8.1. First Animal Experiment

The experiment aimed to evaluate survival rate, tumor incidence, lymphoid organ weight, and lymphocyte proportions in peripheral blood mononuclear cells (PBMCs), the spleen, and thymus. A total of 93 one-day-old chicks were randomly divided into three groups: rRB-1B_Meq (*n* = 35); rRB-1B_Meq71/77 (*n* = 33); and phosphate-buffered saline (PBS, pH 7.4) as the negative control (*n* = 25). Each group was housed separately in individual isolators.

##### In Vivo Kinetics of Recombinant Viruses and T Cell Subset Dynamics

Chickens were inoculated intra-abdominally with 5000 pfu of rRB-1B_Meq (*n* = 20), rRB-1B_Meq71/77 (*n* = 20), or PBS (*n* = 20). At 7, 14, 28, and 35 days post-infection (dpi), five chickens from each group were euthanized under deep general anesthesia using isoflurane inhalation (Zoetis Japan, Tokyo, Japan). Following euthanasia, the blood, spleen, thymus, bursa of Fabricius, and feather tips were collected after recording body weight. The weights of the spleen, thymus, and bursa of Fabricius were measured, and their ratios to body weight were calculated. The spleen and thymus were mechanically dissociated with scissors, homogenized, and passed through 40-μm cell strainers (BD Biosciences, San Jose, CA, USA) to prepare single-cell suspensions, which were washed twice with PBS. Mononuclear cells were isolated from blood and tissue suspensions via density gradient centrifugation using Percoll (GE Healthcare, Chicago, IL, USA) and stored in Cell Banker 1 (Nippon Zenyaku Kogyo Co., Ltd., Fukushima, Japan) for subsequent flow cytometric analysis.

##### Virulence of Recombinant Viruses

To evaluate virulence, disease incidence was monitored over 8 weeks following intra-abdominal inoculation with rRB-1B_Meq (*n* = 15) or rRB-1B_Meq71/77 (*n* = 13). Chickens were observed daily, and those displaying clinical signs, including neurological symptoms or depression, were euthanized and subjected to necropsy for tumor assessment. At 56 dpi, asymptomatic chickens were also euthanized for tumor evaluation: control (*n* = 5), rRB-1B_Meq-infected group *(n* = 8), and rRB-1B_Meq71/77-infected (*n* = 13). Mononuclear cells were isolated from the blood, spleen, thymus, and bursa as described previously.

#### 2.8.2. Second Animal Experiment

A second experiment was conducted to evaluate survival rate, tumor incidence, and histopathological changes. A total of 45 one-day-old chicks were randomly allocated into three groups and housed separately. Chickens were inoculated intra-abdominally with 10,000 pfu of rRB-1B_Meq (*n* = 20), rRB-1B_Meq71/77 (*n* = 20), or PBS (*n* = 5) as a control. Blood samples were collected from five chickens per group at 7, 14, 28, 35, and 49 dpi to monitor viral load kinetics. Clinical signs of MD were monitored daily until 56 dpi. At 56 dpi, all surviving asymptomatic chickens were euthanized for tumor assessment: control (*n* = 5), rRB-1B_Meq-infected (*n* = 10), and rRB-1B_Meq71/77-infected (*n* = 20). Blood was collected to quantify viral loads. For histopathological analysis, the spleen, bursa of Fabricius, and thymus were collected at 56 dpi: control (*n* = 2), rRB-1B_Meq-infected (*n* = 3), and rRB-1B_Meq71/77-infected (*n* = 5).

### 2.9. Histopathological Analysis of rMDV-Infected Chickens

The spleen, thymus, and bursa of Fabricius from experimentally infected chickens were fixed in 10% neutral-buffered formalin (Fujifilm Wako Pure Chemical Corporation, Osaka, Japan Osaka, Japan) for 24 h, followed by paraffin embedding. After deparaffinization, sections were incubated with 0.3% H_2_O_2_ in methanol for 20 min at room temperature (20–26 °C) to block endogenous peroxidase activity. For hematoxylin-eosin (HE) staining, the slides were subsequently counterstained with Mayer’s hematoxylin (Muto Pure Chemicals, Tokyo, Japan). For immunohistochemistry (IHC), to prevent nonspecific antibody binding, the sections were incubated with 5% skim milk solution (Fujifilm Wako Pure Chemical Corporation, Osaka, Japan) for 20 min at room temperature. Supernatants from cloned hybridoma cell cultures containing anti-Meq mAbs [63] were used as primary antibodies. The sections were then incubated with anti-Meq mAbs (1.4 mg/mL) for 20–24 h at 4 °C. Following PBS rinsing, the sections were treated with a horseradish peroxidase polymer-based secondary antibody reagent (Histofine Simple Stain MAX PO [M] kit, Nichirei Bioscience, Tokyo, Japan) for 45 min at room temperature (20–26 °C). Antigen–antibody reactions were visualized using a DAB substrate kit (Nichirei Bioscience, Tokyo, Japan). The slides were counterstained with Mayer’s hematoxylin (Muto Pure Chemicals, Tokyo, Japan) and coverslipped. Immunostaining was observed under a microscope.

### 2.10. Flow Cytometric Analysis

#### 2.10.1. Staining of Mononuclear Cells from Spleen and Thymus

A portion of the single-cell suspension (5 × 10^5^ cells), prepared from the spleen and thymus, was seeded into 96-well round-bottom plates and washed twice with a FACS buffer (PBS supplemented with 1% bovine serum albumin; Sigma-Aldrich, St. Louis, MI, USA). Mononuclear cells were then blocked with PBS containing 10% chicken serum (Thermo Fisher Scientific, Waltham, MA, USA) at 25 °C for 15 min. As shown in the gating strategy (Appendix A), cells were stained for 30 min at 4 °C in the dark with the following antibodies: PerCP-Cyanine5.5 (PerCP-Cy5.5)-conjugated mouse anti-chicken CD3ε monoclonal antibody (mAb), PE-Cyanine7 (PE-Cy7)-conjugated anti-chicken CD4 mAb, fluorescein-5-isothiocyanate (FITC)-conjugated mouse anti-chicken CD8β mAb, and phycoerythrin-conjugated mouse anti-chicken TCRγδ mAb. All antibodies were purchased from Southern Biotech, Birmingham, AL, USA. Dead cells were excluded using Fixable Viability Dye eFluor780 (Thermo Fisher Scientific, Waltham, MA, USA). Following two washes with the FACS buffer, cells were fixed and permeabilized with Cytofix/Cytoperm solution (BD Biosciences, San Jose, CA, USA) for 30 min at 4 °C, then washed three times with a Perm/Wash buffer (BD Biosciences, San Jose, CA, USA). For intracellular staining, cells were incubated for 40 min at 4 °C in the dark with either allophycocyanin (APC)-conjugated mouse anti-Meq mAb or APC-conjugated mouse IgG_1_ isotype control (Southern Biotech San Jose, CA, USA), followed by two washes with the Perm/Wash buffer. The anti-Meq mAb (mouse IgG1 isotype), previously established in [63], was purified using Protein G Sepharose 4 Fast Flow (GE Healthcare, Chicago, IL, USA) and conjugated with APC using the Zenon Mouse IgG1 Labeling Kit (Invitrogen, Waltham, MA, USA). Its specificity was validated in a prior study [56]. Finally, cells were resuspended in 200 μL of the FACS buffer and analyzed on a FACSLyric flow cytometer (BD Biosciences, San Jose, CA, USA). The absolute number of each T cell subset in a proportion of cell suspension was quantified using CountBright™ absolute counting beads (Invitrogen, Waltham, MA, USA) and the following formula:(Number of T cell subset events/Number of bead events) × Number of beads added

The resulting value was used to estimate the total absolute number of each T cell subset in the entire single-cell suspension.

#### 2.10.2. Staining of PBMCs

PBMCs (5 × 10^5^) were seeded into 96-well round-bottom plates and washed twice with a FACS buffer. Cells were blocked with PBS containing 10% chicken serum (Thermo Fisher Scientific, Waltham, MA, USA) at 25 °C for 15 min. As outlined in the gating strategy (Appendix A), cells were stained for 30 min at 4 °C in the dark with the following monoclonal antibodies: PerCP-Cy5.5-conjugated mouse anti-chicken CD3ε, PE-Cy7-conjugated anti-chicken CD4, FITC-conjugated mouse anti-chicken CD8β, PE-conjugated mouse anti-chicken TCRγδ, and APC-conjugated anti-chicken CD8α. All antibodies were from Southern Biotech. Dead cells were stained with Fixable Viability Dye eFluor780 (Thermo Fisher Scientific, Waltham, MA, USA). After two washes with the FACS buffer, cells were analyzed on a FACSLyric flow cytometer (BD Biosciences, San Jose, CA, USA).

### 2.11. Quantification of Viral Loads

Total cellular DNA was extracted from whole blood and from cell suspensions of the spleen, thymus, and bursa of Fabricius (not used for flow cytometric analysis) using the DNeasy Blood and Tissue Kit (Qiagen, Hilden, Germany), according to the manufacturer’s instructions. DNA from feather tips was extracted as described previously [64]. Viral loads in infected chickens were quantified by qPCR targeting the *infected cell protein 4* (*icp4*) gene of MDV. Reactions were conducted using TB Green Premix DimerEraser (Takara Bio Inc., Kyoto, Japan) on a LightCycler 96 System (Roche Diagnostics, Mannheim, Germany). The chicken-*inducible nitric oxide synthase* (*i-nos*) gene was amplified as an internal reference. Viral load values are presented as the ratio of *icp4* to *i-nos*. Primer sequences used for qPCR are listed in Table 4.

### 2.12. Statistical Analyses

All statistical analyses were performed using R Statistical Software (version 4.0.3; R Foundation for Statistical Computing, Vienna, Austria). Transactivation activity was analyzed using Tukey’s multiple comparison test. Multistep growth kinetics were analyzed using the Mann–Whitney U test. Lymphoid organ-to-body weight ratios and T cell subset dynamics were analyzed using Dunn’s test. Tumor incidence was analyzed using Fisher’s exact test. A *p*-value < 0.05 was considered statistically significant.

## 3. Results

### 3.1. Effects of Polymorphisms in CVI988-Meq on Transcriptional Regulation Activity of RB-1B-Meq

A comparison of the amino-acid sequences between RB-1B-Meq and CVI988-Meq revealed polymorphisms at positions 71, 77, and 326 (Table 1). To elucidate the effects of these polymorphisms in CVI988-Meq on Meq-mediated transcriptional regulation, we introduced amino acid substitutions at positions 71, 77, and 326 in RB-1B-Meq—namely, Meq(A71S), Meq(K77E), and Meq(T326I)—as well as a triple mutant containing all three substitutions (Meq(Triple)), and conducted reporter assays using the promoters of Meq target genes. Although all Meq isoforms exerted transrepression effects on the pp38 and pp14 promoters, the transrepression activity of Meq(K77E) and Meq(Triple) was significantly weaker than that of wild-type RB-1B-Meq(Meq(RB-1B)) (Figure 1A,B), suggesting that the amino acid residue at position 77 is primarily responsible for the reduced transrepression observed in CVI988-Meq. Furthermore, although all Meq constructs enhanced transactivation activity on the promoters of viral (meq, icp4, and gb) and host (bcl-2, il-2, and cd30) genes, the Meq isoforms containing amino acid substitutions generally exhibited lower transactivation activity compared to Meq(RB-1B) (Figure 1C–H). Notably, Meq(A71S) and Meq(Triple) showed markedly reduced transactivation activity on all promoters relative to Meq(RB-1B). In contrast, Meq(T326I) showed a milder reduction in transactivation activity on the gb and il-2 promoters compared to the other mutant isoforms. Additionally, Meq(Triple) was the only isoform that showed significantly lower transactivation activity on the cd30 promoter than Meq(RB-1B), suggesting potential synergistic effects among these polymorphisms in CVI988-Meq. Taken together, the findings indicate that polymorphisms in CVI988-Meq diminish the transcriptional regulatory functions of Meq, with the residue at position 71 likely contributing to reduced transactivation activity and the residue at position 77 to diminished transrepressive function.

### 3.2. Generation and Characterization of Recombinant Viruses in Vitro

Because the amino acid substitutions at positions 71 and 77 in the basic region had a greater impact on the transcriptional activity of Meq than the substitution at position 326, we generated an RB-1B-based rMDV encoding Meq with serine and glutamic acid at positions 71 and 77, respectively (Meq71/77), to investigate whether these polymorphisms also affect MDV virulence. We inserted either *RB-1B-meq* or *meq71/77* into the *meq* locus in the TRL region of pRB-1B_ΔIRL_Δmeq (Figure 2A). RFLP analysis confirmed the correct insertion of each *meq* gene into the respective loci in the resulting BAC plasmids, pRB-1B_ΔIRL_Meq and pRB-1B_ΔIRL_Meq71/77 (Appendix A). Additionally, PCR analysis verified the restoration of the IRL region in the recombinant viruses, rRB-1B_Meq and rRB-1B_Meq71/77, reconstituted from each BAC plasmid (Appendix A).

To confirm whether this recombination process affected viral replication ability or *meq* mRNA expression in vitro, we analyzed viral loads and the transcript levels of meq in CEFs infected with rRB-1B_Meq and rRB-1B_Meq71/77. No significant differences in replication kinetics were observed between the two viruses in vitro (Figure 2B). Additionally, RT-qPCR confirmed that the transcript levels of *meq* were not altered by the recombination process (Figure 2C).

### 3.3. Virulence of rMDVs In Vivo

To evaluate the impact of amino acid residues at positions 71 and 77 on MDV virulence and tumor formation, 1-day-old chickens were infected with 5000 pfu of rRB-1B_Meq or rRB-1B_Meq71/77. At 56 dpi, 60.0% of rRB-1B_Meq-infected chickens showed clinical signs such as leg paralysis and torticollis, and 56.7% exhibited solid tumor formation in visceral organs, including those euthanized at 56 dpi (Figure 3A,B). In contrast, none of the chickens in the rRB-1B_Meq71/77-infected group developed clinical signs or solid tumors. To assess whether rRB-1B_Meq71/77 retains any pathogenic potential at a higher dose, 1-day-old chickens were infected with 10,000 pfu of rMDVs. In the rRB-1B_Meq-infected group, 50.0% of chickens developed clinical signs, and 55.0% exhibited solid lymphomas (Figure 3C,D). In contrast, no clinical signs or tumor formation were observed in rRB-1B_Meq71/77-infected chickens, even at the higher dose, indicating that the substitution at positions 71 and 77 markedly reduces the virulence and oncogenicity of MDV.

### 3.4. Changes in Lymphoid Organ Weight in Chickens Infected with rMDVs

Lytic replication of MDV induces extensive apoptosis in the thymus and bursa of Fabricius, resulting in atrophy of these organs [21,65]. In addition, MDV transforms CD4^+^ T cells into lymphoma cells, resulting in their proliferation and subsequent enlargement of the spleens in infected chickens [11]. To evaluate the effects of rMDVs on lymphoid organs, we compared the weight of the thymus, bursa, and spleen at each time point. The thymus weight in the rRB-1B_Meq-infected group tended to be lower than that in the control group at 7 dpi and was significantly lower at 14 dpi (Figure 4A), indicating thymic atrophy. In contrast, although rRB-1B_Meq71/77-infected chickens showed a trend toward lower thymus weight, no significant difference was observed at 7 and 14 dpi compared to the control group (Figure 4A). The bursa weight in rRB-1B_Meq-infected chickens was lower than that in the control group at 14 dpi (Figure 4B); however, no significant differences in bursa weight between the rRB-1B_Meq71/77-infected and control groups were observed throughout the experimental period (Figure 4B). These findings suggest that rRB-1B_Meq71/77 has a reduced potential to induce atrophy of the thymus and bursa compared to rRB-1B_Meq. The spleens of both rMDV-infected groups tended to be larger than those of the control group at 7 dpi (Figure 4C). From 14 dpi onward, the spleen weight in the rRB-1B_Meq-infected group increased compared to the other two groups, and was significantly higher at 56 dpi. In contrast, no differences in spleen weight between the rRB-1B_Meq71/77-infected and control groups were observed after 14 dpi. Thus, transient pathogenic changes in lymphoid tissues were observed in the early phase of infection in rRB-1B_Meq71/77-infected chickens, but no significant changes were noted in the later phase.

### 3.5. Histopathological Analysis of Chickens Infected with rMDVs

Histopathological analysis was performed on chickens at 56 dpi in the second animal experiment (control group: *n* = 2; rRB-1B_Meq71/77-infected group: *n* = 5; rRB-1B_Meq-infected group: *n* = 3). All rRB-1B_Meq-infected chickens exhibited extensive infiltration of Meq^+^ cells throughout the spleen, along with macrophage infiltration in the red pulp. In contrast, although Meq^+^ cells were detected in the spleens of two out of five rRB-1B_Meq71/77-infected chickens, no proliferative tumor lesions were observed in this group (Figure 5A,B, and Table 5). The thymus and bursa of Fabricius of all rRB-1B_Meq-infected chickens also exhibited infiltration of Meq^+^ cells. In contrast, although Meq^+^ cells were also detected in the thymus and bursa of two out of five rRB-1B_Meq71/77-infected chickens, no infiltration of proliferative tumor cells was observed in the thymus or bursa of this group (Figure 5C,D, and Table 5).

### 3.6. In Vivo Replication of rMDVs

To investigate the replication kinetics of rMDVs in vivo, we quantified the viral loads in whole blood, spleen, thymus, and feather tips using qPCR. In the first and second experiments, whole blood from rRB-1B_Meq71/77-infected chickens exhibited significantly lower viral loads than that of rRB-1B_Meq-infected chickens in the later phase of infection (Figure 6A,B). Additionally, viral loads in the spleen and thymus were significantly lower in rRB-1B_Meq71/77-infected chickens after 14 dpi compared to those in rRB-1B_Meq-infected chickens (Figure 6C,D), and viral load in feather tips of rRB-1B_Meq71/77-infected chickens was significantly lower at 56 dpi (Figure 6E). These findings suggest that rRB-1B_Meq71/77 exhibits lower replication capacity in vivo compared to rRB-1B_Meq, or that the proliferative capacity of cells infected with rRB-1B_Meq71/77 is lower than that of cells infected with rRB-1B_Meq.

### 3.7. Dynamics of CD4^+^ T and Meq^+^ Cells in the Spleens of Chickens Infected with rMDVs

We investigated the dynamics of T cell subsets by analyzing spleens from chickens challenged with rMDVs at each time point in the first experiment. The absolute number of CD4^+^ T cells—primary targets of MDV transformation—was significantly higher in rRB-1B_Meq-infected chickens compared to the control group after 28 dpi (Figure 7A,B). Similarly, the absolute number of CD4^+^ T cells in rRB-1B_Meq71/77-infected chickens tended to increase at 28 dpi, reaching significantly higher levels than in the control group at 35 dpi; however, by 56 dpi, the number of CD4^+^ T cells returned to levels comparable to the control group. Notably, the absolute number of CD4^+^ T cells in rRB-1B_Meq71/77-infected chickens was consistently lower than in the rRB-1B_Meq-infected group from 14 dpi onward, with a statistically significant difference at 56 dpi.

To evaluate the dynamics of rMDV-infected cells, we examined the proportion of Meq^+^ cells within the CD4^+^ T cell population. In rRB-1B_Meq-infected chickens, Meq^+^CD4^+^ T cells were detected in all individuals, and the proportion of Meq^+^ cells among CD4^+^ T cells and the absolute number of Meq^+^CD4^+^ T cells progressively increased throughout the experimental period (Figure 7C,D). In contrast, in rRB-1B_Meq71/77-infected chickens, the proportion of Meq^+^ cells among CD4^+^ T cells averaged 7.1% at 7 dpi but gradually declined to below 1% after 28 dpi (Figure 7C). The proportion was significantly lower than in rRB-1B_Meq-infected chickens at all time points. Furthermore, the absolute number of Meq^+^CD4^+^ T cells did not increase over time in rRB-1B_Meq71/77-infected chickens, and remained significantly lower than in rRB-1B_Meq-infected chickens from 14 dpi onward (Figure 7D).

### 3.8. Dynamics of CD8^+^ T and γδ T Cells in the Spleens of Chickens Infected with rMDVs

We analyzed the dynamics of CD8^+^ T cells, which play a crucial role in cellular immunity against MDV infection and tumor cells [66]. In rRB-1B_Meq-infected chickens, the absolute number of CD8αβ^+^ T cells tended to increase at 7 dpi, though the difference from the control group was not statistically significant. The absolute number of CD8αβ^+^ T cells in rRB-1B_Meq-infected chickens was significantly higher than that in the control group from 14 to 35 dpi (Figure 7E,F). In contrast, rRB-1B_Meq71/77-infected chickens exhibited a significantly higher number of CD8αβ^+^ T cells than the control group at 7 dpi. Additionally, at 28 and 35 dpi, the number of CD8αβ^+^ T cells in rRB-1B_Meq71/77-infected chickens remained significantly higher than in the control group, but tended to be lower than that in rRB-1B_Meq-infected chickens.

Next, we examined the dynamics of γδ T cells, which are known to play protective roles against MD in chickens vaccinated with the CVI988 strain [31,32]. In rRB-1B_Meq-infected chickens, similar to the trend observed in the number of CD8αβ^+^ T cells, the number of γδ T cells slightly increased at 28 and 35 dpi compared to the control group. In contrast, in rRB-1B_Meq71/77-infected chickens, the number of γδ T cells was significantly higher than that in the control group at 7 dpi. Thereafter, the number of γδ T cells in rRB-1B_Meq71/77-infected chickens remained comparable to that of the control group.

### 3.9. Dynamics of T Cell Subsets in the Thymus of Chickens Infected with rMDVs

We analyzed the dynamics of T cell subsets in the thymus. In the rRB-1B_Meq71/77-infected group, the absolute number of CD4^+^ T cells (TCRγδ^−^CD3^+^CD4^+^) was significantly decreased at 7 dpi compared to that in the control group (Figure 8A,B). A similar trend was also observed in the rRB-1B_Meq-infected group, suggesting cytolytic infection in CD4^+^ T cells in both rMDV-infected groups. In contrast, after 14 dpi, both rMDV-infected groups showed a trend toward an increase in the numbers of CD4^+^ T cells compared to the control group, with statistically significant increases observed at some time points. Furthermore, the proportion of Meq^+^ cells within the CD4^+^ T cell population and the absolute number of Meq^+^ CD4^+^ T cells in rRB-1B_Meq71/77-infected chickens were not increased and were lower than those in rRB-1B_Meq-infected chickens from 14 dpi (Figure 8C,D). These findings suggest that infected T cells did not expand in the thymus of the rRB-1B_Meq71/77-infected group, despite an overall increase in the number of CD4^+^ T cells in this group. The number of CD8αβ^+^ T cells (TCRγδ^−^CD3^+^CD8β^+^) exhibited a similar pattern to the number of CD4^+^ T cells (Figure 8E,F). Although the number of CD8αβ^+^ T cells in both rMDV-infected groups slightly decreased at 7 dpi, it subsequently tended to increase compared to the control group after 14 dpi, with statistically significant increases observed at 28, 35, and 56 dpi. In contrast, the absolute number of γδ T cells (TCRγδ^+^CD3^+^) in the thymus decreased at 7dpi in both rMDV-infected groups, with a significantly lower count observed in the rRB-1B_Meq-infected group compared to the control group. Thereafter, the number of γδ T cells remained consistently lower than that of the control group throughout the experimental period (Figure 8G,H).

### 3.10. Dynamics of T Cell Subsets in PBMCs of Chickens Infected with rMDVs

We also investigated the dynamics of T cell subsets in PBMCs of chickens challenged with rMDVs in the first experiment. Similarly to the spleen, the proportion of CD4^+^ T cells within the T cell population in both rMDV-infected groups tended to be higher than that of the control group at 7 dpi (Figure 9A). However, except for a significant increase in CD4^+^ T cells observed in the rRB-1B_Meq-infected group at 35 dpi, no marked changes in CD4^+^ T cell numbers were detected.

To examine the response of CD8 T cell subsets in infected chickens, we employed a gating strategy using the anti-chicken CD8α antibody, as shown in Appendix A. In PBMCs from both rMDV-infected chickens, the proportion of CD8α^+^ T cells (TCRγδ^−^CD3^+^CD8α^+^) increased at 7 dpi compared to that in the control group (Figure 9B). Although a transient decrease was observed in the rRB-1B_Meq71/77-infected group at 28 dpi, the proportion of CD8α^+^ T cells tended to increase from 35 dpi in both rMDV-infected groups compared with the control group. However, no significant difference was observed between the two rMDV-infected groups throughout the experimental period. Given that the proportion of CD8αα^+^ T cells (TCRγδ^−^CD3^+^CD8α^+^CD8β^-^) increases during the proliferation phase in MDV-infected chickens [36], we also investigated the proportion of CD8αα^+^ T cells within CD8α^+^ T cell populations. The proportion of CD8αα^+^ T cells tended to be higher in rRB-1B_Meq-infected chickens throughout the experimental period (Figure 9C). Furthermore, in rRB-1B_Meq71/77-infected chickens, the proportion of CD8αα^+^ T cells within the CD8α^+^ T cell population was significantly higher than that in the control group at 7, 14, and 35 dpi. However, the proportion of CD8αα^+^ T cells was comparable to that of the control group at 49 and 56 dpi.

In PBMCs from rRB-1B_Meq-infected chickens, the proportion of γδ T cells tended to decrease after 14 dpi, with a significant reduction observed at 56 dpi compared to the other two groups (Figure 9D). In contrast, the proportion of γδ T cells in the rRB-1B_Meq71/77-infected group remained comparable to that of the control group and was significantly higher than in the rRB-1B_Meq group at 14 and 56 dpi. Previous studies reported that infection with the RB-1B strain increases CD8α^+^ γδ T cells (TCRγδ^+^CD3^+^CD8α^+^) during the transformation phase [67], suggesting a role for this subset in responding to infected or transformed cells. In this study, the proportion of CD8α^+^ γδ T cells among γδ T cells was elevated in both rMDV-infected groups compared to the control group at 14 and 28 dpi (Figure 9E). Notably, the proportion of CD8α^+^ γδ T cells significantly increased in rRB-1B_Meq-infected chickens compared to the control group at 56 dpi, whereas the rRB-1B_Meq71/77-infected group showed a declining trend after 35 dpi.

Taken together, rRB-1B_Meq71/77-infected chickens did not exhibit an increase in Meq^+^CD4^+^ T cells within PBMCs, as observed in rRB-1B_Meq-infected chickens. This suggests a lack of expansion of infected or transformed cells in the rRB-1B_Meq71/77 group. Furthermore, although both rMDV-infected groups showed increases in CD8α^+^ T and γδ T cells during the early phase of infection, the proportions of CD8α^+^ T and CD8α^+^ γδ T cells—likely responding to MDV-infected cells—gradually declined during the late phase in rRB-1B_Meq71/77-infected chickens, returning to levels comparable to those in the control group.

## 4. Discussion

In this study, we investigated the effects of amino acid polymorphisms in the basic region of CVI988-Meq on Meq-mediated transcriptional regulation and MDV virulence. Introducing these polymorphisms into RB-1B-Meq significantly reduced transcriptional regulatory activity. Moreover, chickens infected with rRB-1B_Meq71/77 showed no clinical signs of MD or gross tumors and exhibited a marked reduction in Meq^+^ cells in the spleen and thymus, suggesting that rRB-1B_Meq71/77 may fail to establish latent infection effectively or promote transformation of infected T cells.

The introduction of amino acid polymorphisms in CVI988-Meq markedly impaired Meq-mediated transcriptional repression on the pp38 and pp14 promoters. Since enforced expression of *pp38* using CRISPR activation promotes lytic infection by upregulating other lytic genes such as *icp4* and *pp14* [68], the Meq isoform carrying CVI988-Meq polymorphisms may be less capable of establishing latency as a result of reduced suppression of *pp38* expression. These polymorphisms also decreased the transactivation activity of Meq across all promoters tested, including those associated with transformation, such as *bcl-2*, *cd30*, and *meq.* This supports the hypothesis that CVI988-Meq polymorphisms contribute to the lower transformation potential and the low virulence of the CVI988 strain. Additionally, mutations at positions 71 and 77 in the basic region of RB-1B-Meq impaired transactivation activity more significantly than the mutation at position 326. Since the basic region mediates genomic DNA binding, polymorphisms in this region may weaken the affinity for target promoters. Serine at position 42 in Meq can be phosphorylated by cyclin-dependent kinase 2, which reduces its DNA-binding affinity [69]. Notably, the amino acid residue at position 71 in CVI988-Meq is serine, which is potentially phosphorylated. In addition, glutamic acid at position 77 in CVI988-Meq carries a strong negative charge, similar to phosphorylated residues, and is frequently used as a phosphomimetic residue [70,71], suggesting that it may influence protein behavior by mimicking phosphorylation. As DNA is also negatively charged, the presence of a phosphorylated serine at position 71 and a negatively charged glutamic acid at position 77 may lead to electrostatic repulsion between Meq and genomic DNA, thereby reducing its binding affinity and transactivation activity. Therefore, further studies are needed to investigate the impact of Meq phosphorylation on its transcriptional regulatory functions and virulence. Additionally, these charge alterations may affect protein–protein interactions. Meq interacts with host proteins such as the C-terminal binding protein and p53 through its basic region [41,72]; therefore, it is necessary to investigate whether the polymorphisms in the basic region affect Meq’s interactions with host proteins using co-immunoprecipitation and proximity-dependent biotin labeling.

A recombinant Md5 strain expressing a chimeric Meq composed of the N-terminal domain of CVI988-Meq and the C-terminal domain of Md5-Meq (rMd5-CVI/Md5-Meq) exhibited significantly reduced mortality compared to the parental Md5 strain [49]. These findings highlight the critical role of the N-terminal domain of Meq, particularly the basic region, in MDV virulence. In the present study, introducing only two amino acid polymorphisms at positions 71 and 77 of CVI988-Meq into RB-1B-Meq markedly diminished the virulence of RB-1B. Consistent with our findings, a previous study showed that recombinant RB-1B expressing wild-type CVI988-Meq completely lost virulence [48]. However, rMd5-CVI/Md5-Meq still caused MD in 37% of infected chickens, despite harboring the same polymorphisms at positions 71 and 77 as CVI988-Meq [49]. These observations suggest that RB-1B virulence appears to be highly dependent on the amino acid sequence of Meq, whereas additional viral factors may contribute to Md5 virulence. Thus, genetic differences between RB-1B and Md5 may influence the different outcomes in each rMDV-infected chicken. Additionally, amino acid polymorphisms in other domains of Meq, including the transactivation domain, may contribute to virulence. Notably, highly virulent strains such as N and 648A share polymorphisms where the second prolines in the PPPP motifs of the transactivation domain are substituted with alanine or glutamine [43]. Md5 also harbors a proline-to-alanine substitution at position 217 (P217A) in this motif, whereas RB-1B does not. In our previous study, among the three amino acid differences in Meq between RB-1B and Md5, only the P217A substitution significantly enhanced the transactivation activity of Meq [73]. Therefore, the alanine at position 217 in rMd5-CVI/Md5-Meq may partially contribute to its residual virulence.

In the present study, rRB-1B_Meq71/77-infected chickens did not exhibit tumor formation, although a progressive increase in Meq^+^ cells was observed in the spleens and thymuses of rRB-1B_Meq-infected chickens. Furthermore, no histopathological changes were observed in rRB-1B_Meq71/77-infected chickens. These findings imply that the transformation potential of rRB-1B_Meq71/77 was markedly reduced as a result of the amino acid substitutions at positions 71 and 77. Notably, the proportion of Meq^+^ CD4^+^ T cells in the spleens of rRB-1B_Meq71/77-infected chickens was 7.1% at 7 dpi, but declined to below 1% after 28 dpi. In the spleens of these infected chickens, the numbers of CD8αβ^+^ and γδ T cells were significantly elevated at 7 dpi, and the viral load gradually decreased over time. These data suggest that a robust cellular immune response was elicited in the early phase of infection, resulting in efficient clearance of infected cells. The basic region of Meq is responsible for binding to genomic DNA [37], and mutations in this region may alter the repertoire of Meq-regulated genes. Indeed, rMDVs encoding different Meq variants exhibit variations in the cellular composition of tumor tissues [74], suggesting that Meq sequence variation may influence the expression of genes, including cytokines and chemokines, within tumor cells. Therefore, future studies analyzing the transcriptome of infected cells using approaches such as single-cell RNA sequencing or enrichment of infected cells via reporter viruses will be essential to elucidate the underlying molecular mechanisms.

In the spleens and PBMCs of rMDV-infected chickens, increases in CD8α^+^ T and γδ T cells were observed during early infection. Notably, the proportions of CD8αα^+^ T cells within the CD8α^+^ T population and CD8α^+^ γδ T cells within the γδ T cell population were also elevated in PBMCs during early infection in both infected groups. These subsets have previously been reported to expand during the late phase of RB-1B infection [34,67], suggesting they may respond to MDV-infected or transformed cells. In the present study, although these populations remained elevated during the late phase of infection in the rRB-1B_Meq-infected group, they gradually declined in the rRB-1B_Meq71/77-infected group, returning to levels comparable to those of the control group. This decline, along with the reduction in Meq^+^CD4^+^ T cells, suggests that disease progression was suppressed in rRB-1B_Meq71/77-infected chickens, potentially reflecting immune-mediated clearance and recovery.

MDV induces apoptosis in T cells during early infection, leading to thymic atrophy [21,65]. In this study, the thymus weight of both infected groups tended to be lower than that of the control group during early infection, indicating thymic atrophy associated with cytolytic infection. Consistently, the absolute numbers of CD4^+^ T, CD8αβ^+^ T, and γδ T cells in the thymus of both infected groups decreased at 7 dpi. Although the number of γδ T cells remained consistently lower than that in the control group throughout the observation period, the numbers of CD4^+^ and CD8αβ^+^ T cells increased after 14 dpi in both infected groups. In the rRB-1B_Meq-infected group, this increase in CD4^+^ T cells coincided with an expansion of Meq^+^ cells, implying the proliferation of MDV-transformed cells. However, in rRB-1B_Meq71/77-infected chickens, no such increase in Meq^+^ cells was observed, suggesting that early cytolytic infection may influence the subsequent cellular composition of the thymus. Further investigation is warranted to elucidate the impact of MDV infection on immune responses in the thymus and on T cell differentiation.

Susceptibility to MD is influenced by genetic background and the presence of maternal antibodies. Commercial chickens typically possess maternal antibodies against MDV, which can delay disease onset; therefore, MD-susceptible SPF inbred lines, such as line P or line 7, are generally preferred for evaluating MDV virulence. However, in the present study, experimental infections were conducted using commercial chickens because of the limited availability of these inbred lines in Japan. Thymic and bursal atrophy was observed in both rMDV-infected groups, indicating that early cytolytic infection was not completely masked by maternal antibodies. Nevertheless, it should be noted that the magnitude of early cytolytic infection may have been underestimated under these experimental conditions.

In this study, chickens were inoculated with 5000 or 10,000 pfu of rMDVs, which exceeds the viral doses typically used in MDV pathotyping assays. These higher doses were chosen based on previous studies indicating their suitability for evaluating rMDV virulence over an 8-week experimental period [44,48]. Additionally, since rRB-1B_Meq71/77 carries amino acid polymorphisms shared with the CVI988 strain, we anticipated reduced virulence, and therefore employed higher viral doses to avoid underestimating its pathogenic potential as a result of insufficient challenge.

The present study demonstrates that introducing only two amino acid polymorphisms found in the basic region of CVI988-Meq into RB-1B-Meq markedly reduces the virulence and tumorigenicity of MDV. These findings highlight the pivotal role of the amino acid sequence within the basic region in determining MDV virulence. Notably, low-virulence MDV strains isolated in the U.S. during the 1960s, such as CU-2 and Jm, as well as an ancient MDV strain recovered from 19th-century chicken skeletal remains, possess a serine residue at position 71 of Meq [43,75]. This suggests that the S allele at position 71 may represent the ancestral state, whereas the A allele likely emerged more recently in association with increasing MDV virulence over the past century. Accordingly, further investigation of other polymorphisms in the basic region—such as 71A/77E/80D identified in recent isolates from Asia, Europe, and Africa [76,77,78,79,80], or 71A/77A/80S observed in highly virulent Australian strains [81]—may provide insights into the molecular mechanisms underlying virulence evolution in field strains. Furthermore, as the CVI988 vaccine strain shares the same sequence in the basic region as rRB-1B_Meq71/77, our findings provide mechanistic insight into the low virulence of CVI988, and may contribute to the design of more effective vaccines.

## Figures and Tables

**Figure 1 viruses-17-00907-f001:**
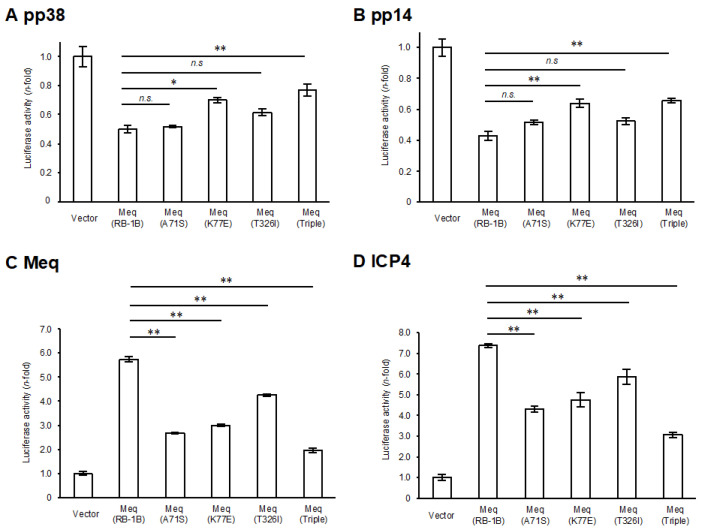
Effects of polymorphisms in CVI988-Meq on transcriptional regulation by Meq. (**A**,**B**) Transrepression effects of Meq with polymorphisms found in CVI988-Meq. The transrepression effects of RB-1B-Meq, Meq(A71S), Meq(K77E), Meq(T326I), and Meq(Triple) on (**A**) the pp38 promoter and (**B**) the pp14 promoter-driven luciferase activities were compared. DF-1 cells in each well were transfected with 300 ng of expression plasmids for each Meq isoform, 500 ng of reporter plasmid, and 5 ng of control pRL-TK plasmid. Luciferase activities were analyzed 24 h post-transfection. Firefly luciferase activity was expressed relative to the mean basal activity in the presence of pCI-neo, after normalization to *Renilla* luciferase activity. (**C**–**H**) Transactivation activities of Meq with polymorphisms in CVI988-Meq. The transactivation activities of RB-1B-Meq, Meq(A71S), Meq(K77E), Meq(T326I), and Meq(Triple) on (**C**) the meq promoter, (**D**) the icp4 promoter, (**E**) the gb promoter, (**F**) the bcl-2 promoter, (**G**) the il-2 promoter, and (**H**) the cd30 promoter-driven luciferase activities were compared. DF-1 cells in each well were transfected with 300 ng of expression plasmid for each Meq isoform, 200 ng of c-Jun expression plasmid, 500 ng of reporter plasmid, and 5 ng of control pRL-TK plasmid. The absolute luciferase activity values for the pCI-neo baseline (in relative light units, RLU) were as follows: A, 47.6; B, 11.1; C, 0.13; D, 0.76; E, 1.61; F, 0.40; G, 0.087; and H, 0.89. Error bars indicate standard deviations. Three independent experiments were performed in triplicate (** *p* < 0.01, * *p* < 0.05; Tukey’s multiple comparison test), n.s.: not significant.

**Figure 2 viruses-17-00907-f002:**
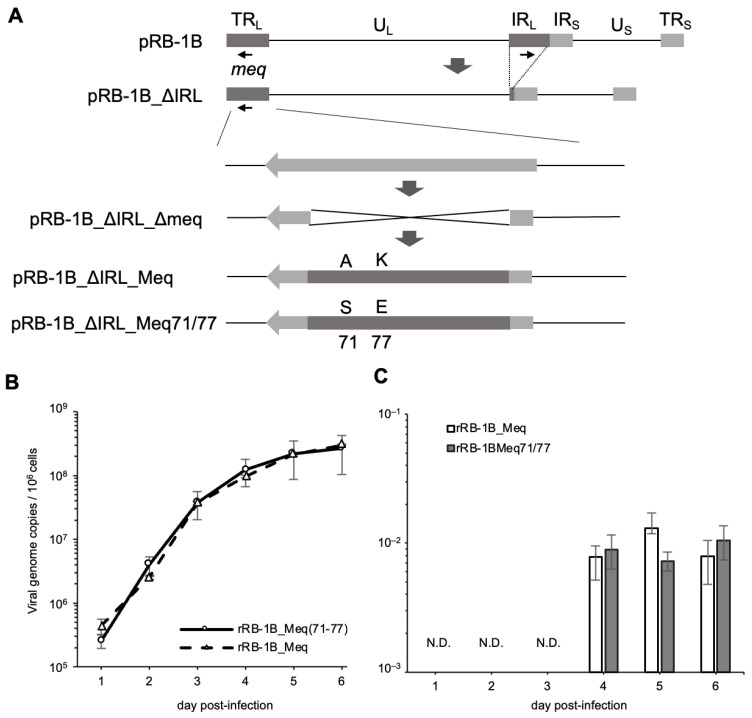
Reconstitution of recombinant Marek’s disease viruses (rMDVs) and their characterization in vitro. (**A**) Schematic diagrams of the constructs with the RB-1B genome in this study. In the RB-1B genome cloned as a bacterial artificial chromosome (BAC) plasmid (pRB-1B), most of the internal repeat long (IRL) regions were deleted in this plasmid, designated as pRB-1B_ΔIRL, and used for mutagenesis. RB-1B-*meq* or *meq*71/77 was inserted into the *meq* locus in the terminal repeat long (TRL) of pRB-1B_ΔIRL with a partial deletion in the *meq* gene (pRB-1B_ΔIRL_Δmeq) by two-step Red-mediated mutagenesis. (**B**) Chicken embryo fibroblasts (CEFs) were infected with 50 plaque-forming units (pfu) of rMDVs. The infected cells were collected daily for 6 days. The viral loads in the infected cells were analyzed by quantitative PCR (qPCR). (**C**) CEFs were infected with 50 pfu of recombinant viruses. Infected cells were collected daily for 5 days. mRNA expression of each Meq isoform in CEFs infected with rRB-1B_Meq and rRB-1B_Meq71/77 was confirmed by reverse transcription-qPCR. The growth and expression kinetics among the groups were analyzed using the Mann–Whitney U test. The experiments were conducted independently in triplicate.

**Figure 3 viruses-17-00907-f003:**
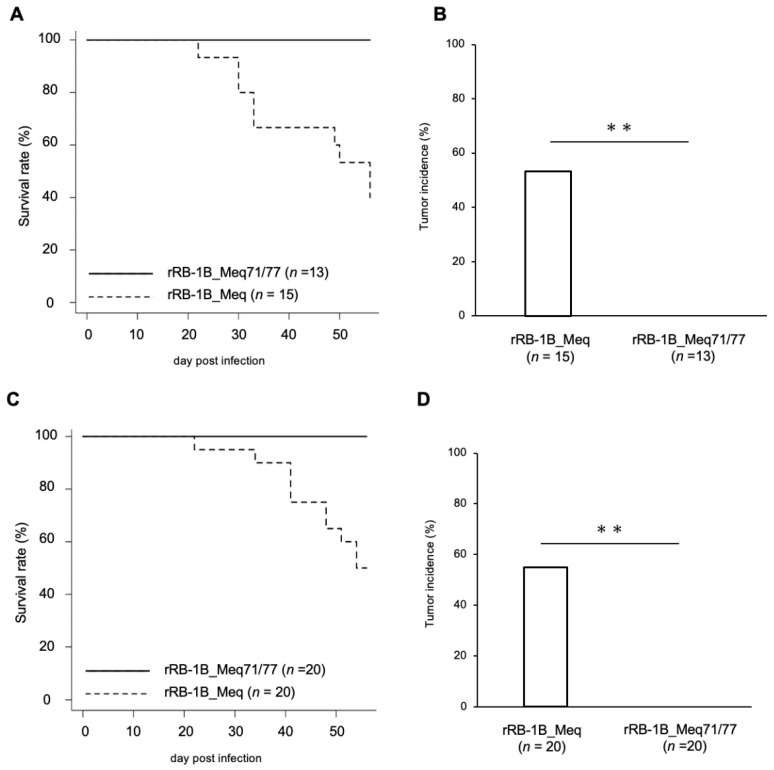
Mortality and tumor incidence in chickens infected with rMDVs. (**A**,**C**) Survival rate in chickens infected with rMDVs in (**A**) the first and (**C**) the second experimental infections. Survival differences between groups were analyzed using the log-rank test; *p*-values were <0.01 in both experiments. (**B**,**D**) Tumor incidence in chickens infected with rMDVs throughout the experimental period, including those euthanized at 56 dpi. Asterisks indicate significant differences (** *p* < 0.01; Fisher’s exact test).

**Figure 4 viruses-17-00907-f004:**
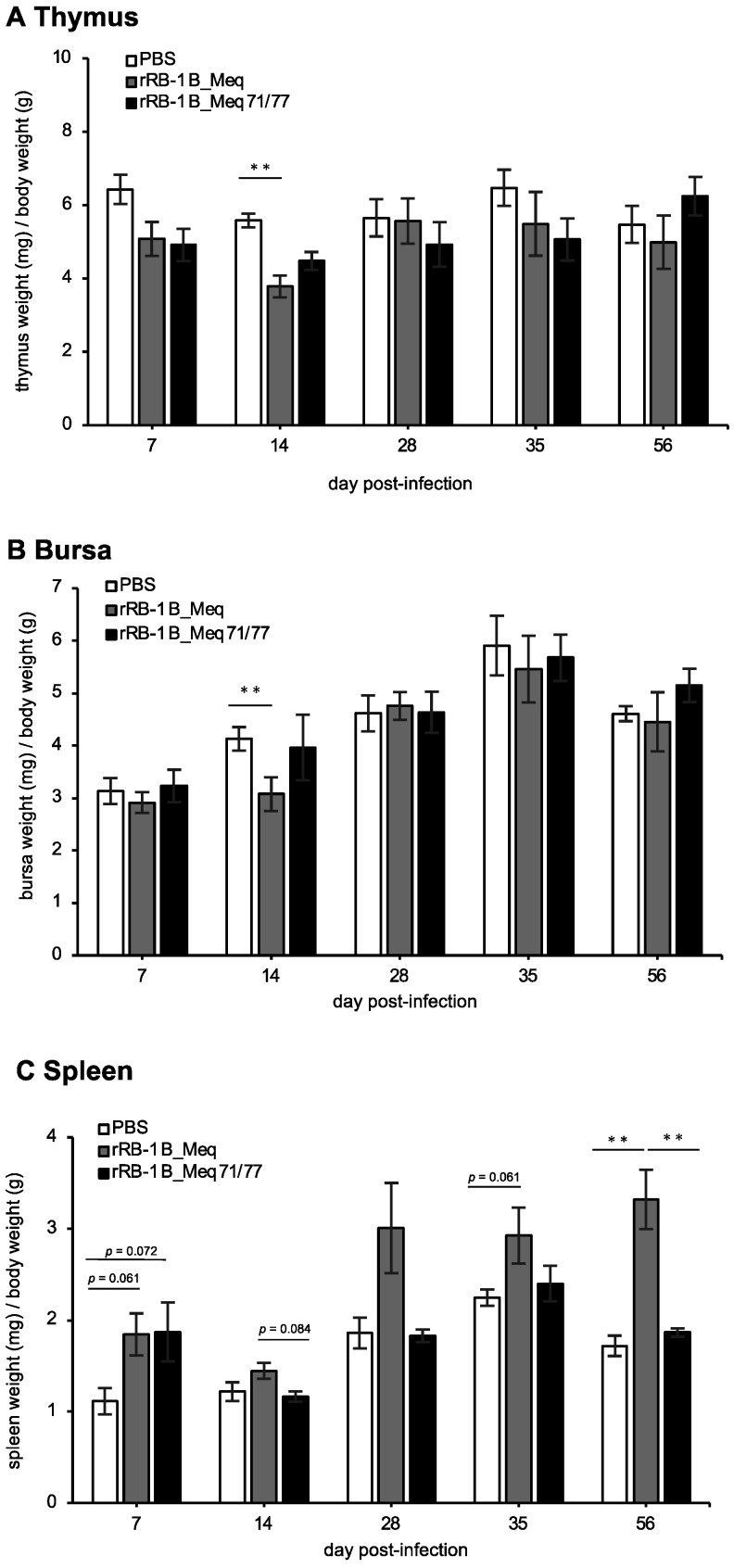
Change in lymphoid organ weight in chickens infected with rMDVs. The dynamics of lymphoid organ-to-body weight ratios for the (**A**) spleen, (**B**) thymus, and (**C**) bursa of Fabricius in chickens infected with rRB-1B-Meq or rRB-1B-Meq71/77 were analyzed at each time point throughout the experimental period. The spleens, thymuses, and bursas of Fabricius were collected from five chickens per group at 7, 14, 28, and 35 dpi, and from all remaining chickens (uninfected control group: *n* = 5, rRB-1B_Meq-infected group: *n* = 13, rRB-1B_Meq71/77-infected group: *n* = 8) at the termination of the experiment at 56 dpi in the first animal experiment. Statistical significance was determined using Dunn’s test (** *p* < 0.01).

**Figure 5 viruses-17-00907-f005:**
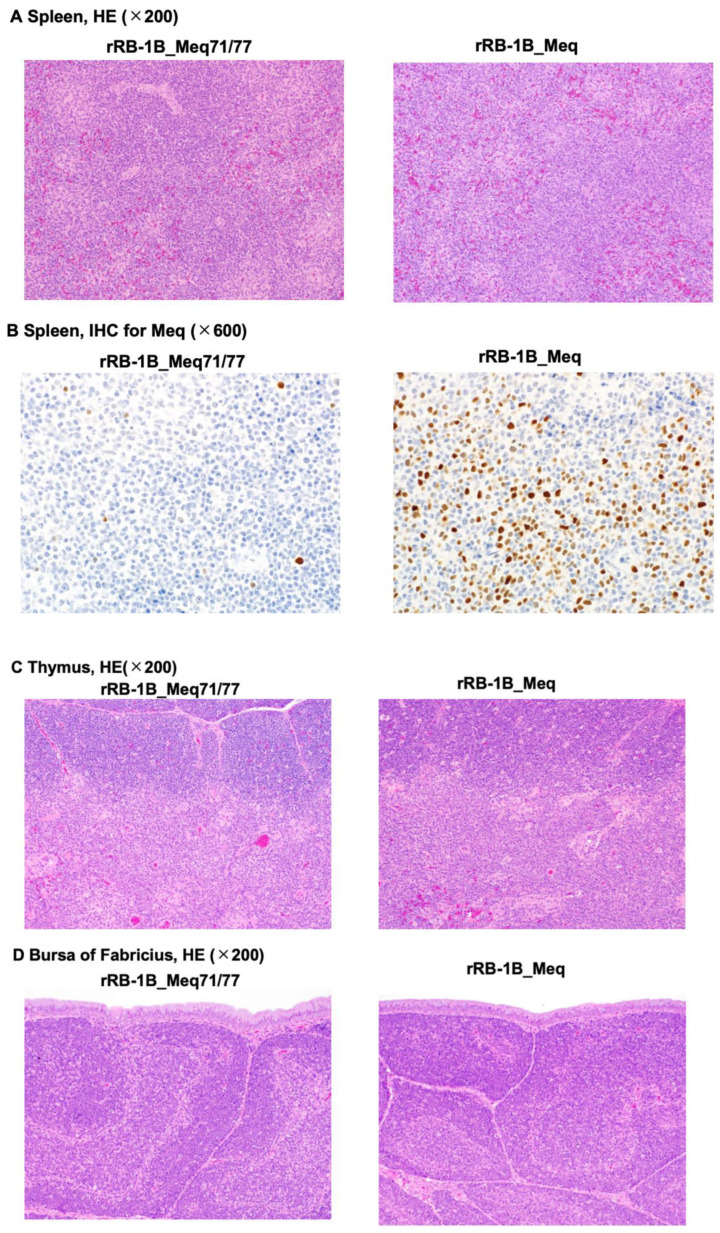
Histopathological analysis of chickens infected with rMDVs. Hematoxylin-eosin staining was performed on the spleen (**A**), thymus (**C**), and bursa of Fabricius (**D**) from chickens in the control, rRB-1B_Meq77/80-infected, and rRB-1B_Meq-infected groups. (**B**) Immunohistochemical staining for Meq in the spleen of chickens infected with either rRB-1B_Meq or rRB-1B_Meq71/77.

**Figure 6 viruses-17-00907-f006:**
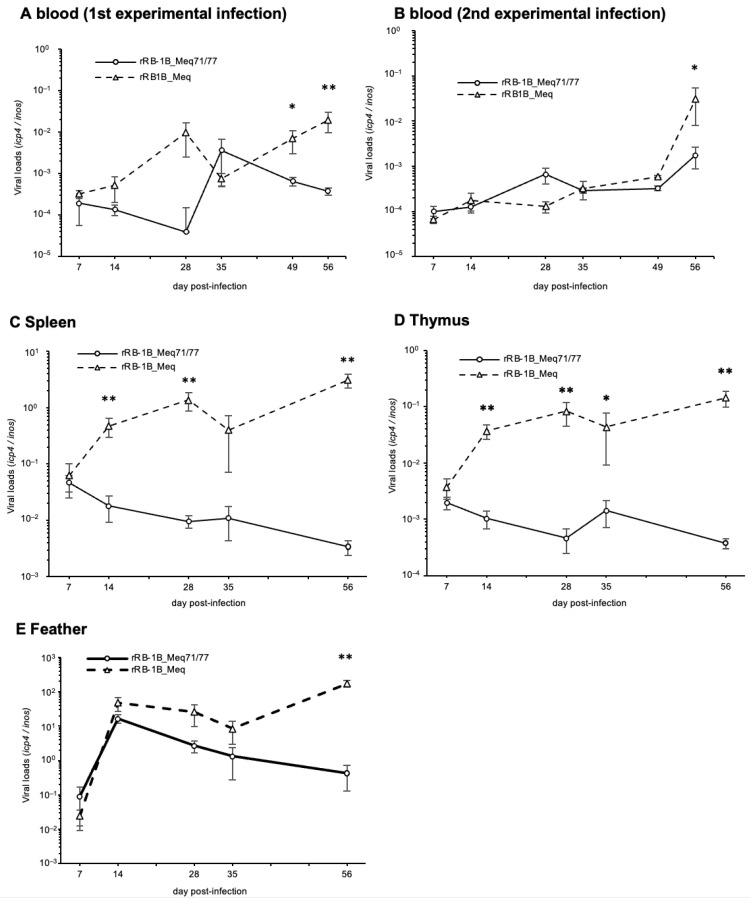
Replication of recombinant Marek’s disease viruses in vivo. The viral loads in the (**A**) whole blood during the first experimental infection, (**B**) whole blood during the second experimental infection, (**C**) spleens, (**D**) thymus, and (**E**) feather tips of chickens infected with rRB-1B_Meq or rRB1B_Meq71/77 were determined by qPCR. Asterisks indicate significant differences (** *p* < 0.01, * *p* < 0.05; Mann–Whitney U test).

**Figure 7 viruses-17-00907-f007:**
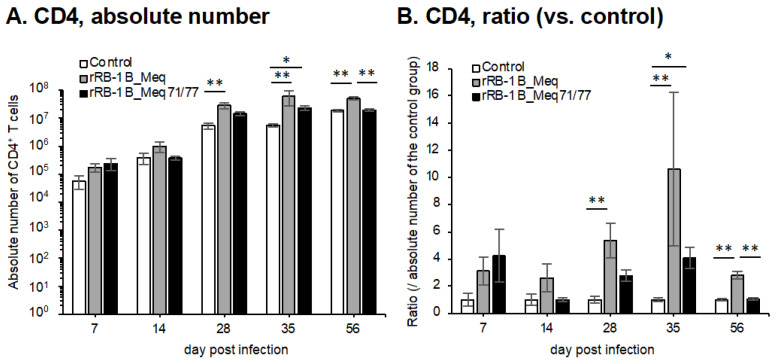
Dynamics of T cell subsets in the spleen of chickens infected with rMDVs. The dynamics of CD4^+^ cells, Meq^+^ cells, CD8^+^ cells, and γδ T cells were analyzed from mononuclear cells isolated from the spleens of chickens infected with rRB-1B-Meq or rRB-1B-Meq71/77 at various time points during the experimental period. The absolute numbers of (**A**) CD4^+^ T cells, (**D**) Meq^+^ cells, (**E**) CD8αβ^+^ T cells, and (**G**) γδTCR^+^ T cells in the spleen were analyzed. The ratios of absolute numbers of (**B**) CD4^+^ T cells, (**F**) CD8αβ^+^ T cells, and (**H**) γδTCR^+^ T cells relative to the control group were also analyzed. (**C**) The percentages of Meq^+^ cells within the CD4^+^ T cell population were determined. Asterisks indicate significant differences (* *p* < 0.05, ** *p* < 0.01; Dunn’s test for (**A**,**B**,**E**–**H**); Mann–Whitney U test for (**C**,**D**)).

**Figure 8 viruses-17-00907-f008:**
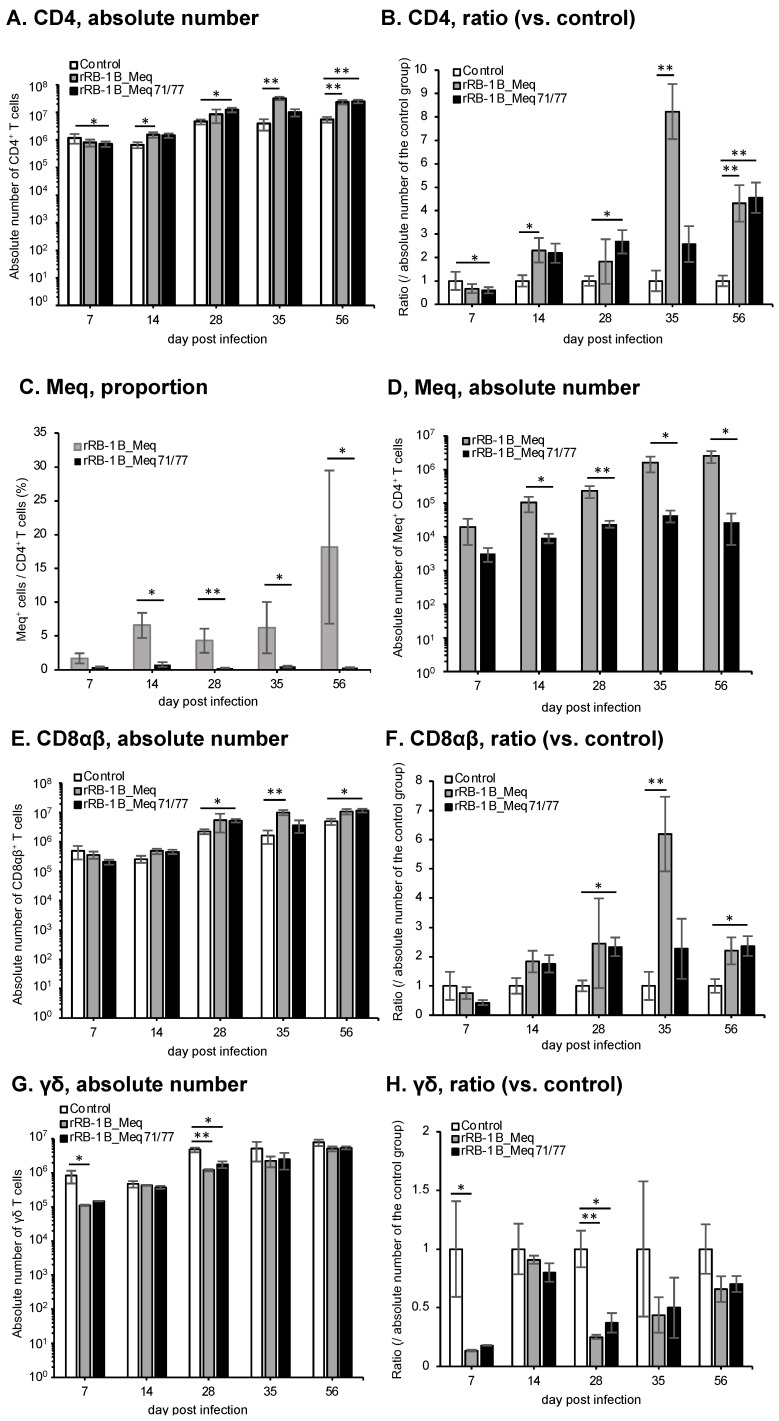
Dynamics of T cell subsets in the thymus of chickens infected with rMDVs. The figure illustrates the dynamics of CD4^+^ cells, Meq^+^ cells, CD8^+^ cells, and γδ T cells derived from mononuclear cells isolated from the thymus of chickens infected with either rRB-1B-Meq or rRB-1B-Meq71/77 at various time points throughout the experimental period. Absolute numbers of (**A**) CD4^+^ T cells, (**D**) Meq^+^ cells, (**E**) CD8αβ^+^ T cells, and (**G**) γδTCR^+^ T cells in the thymus were analyzed. The ratios of absolute numbers of (**B**) CD4^+^ T cells, (**F**) CD8αβ^+^ T cells, and (**H**) γδTCR^+^ T cells in the thymus relative to the control group were analyzed. (**C**) The percentages of Meq^+^ cells in the CD4^+^ T cell population in the thymus were analyzed. Asterisks indicate significant differences (* *p* < 0.05, ** *p* < 0.01; Dunn’s test for (**A**,**B**,**E**–**H**); Mann–Whitney U test for (**C**,**D**)).

**Figure 9 viruses-17-00907-f009:**
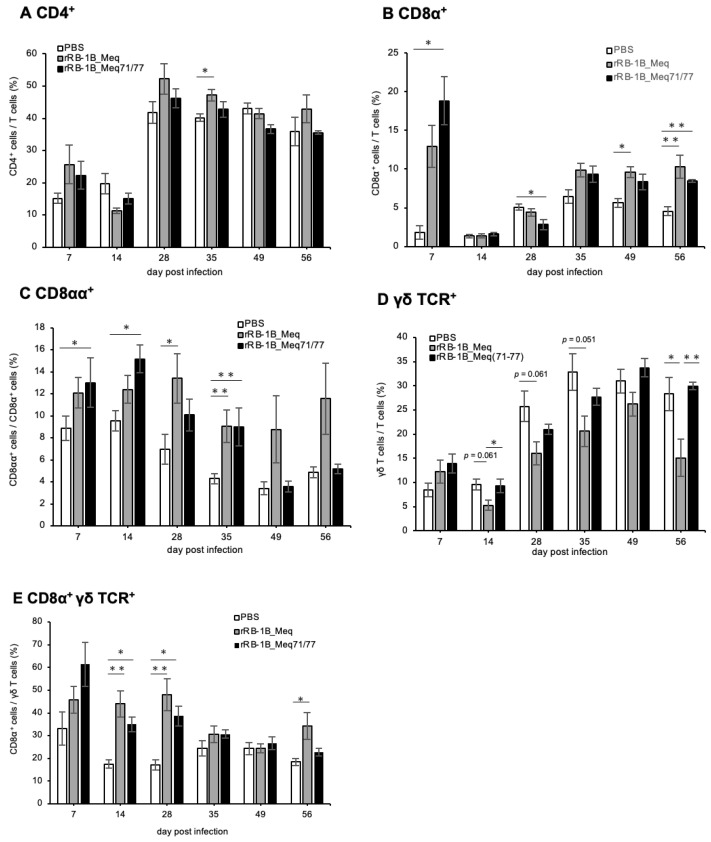
Dynamics of T cell subsets in the peripheral blood mononuclear cells (PBMCs) of chickens infected with rMDVs. This figure depicts the dynamics of CD4^+^ cells, Meq^+^ cells, CD8^+^ cells, and γδ T cells derived from PBMCs of chickens infected with rRB-1B-Meq or rRB-1B-Meq71/77 at various time points during the experimental period. The proportions of (**A**) CD4^+^ T cells within total T cells, (**B**) CD8α^+^ T cells within total T cells, (**C**) CD8αα^+^ T cells within CD8α^+^ T cells, (**D**) γδTCR^+^ T cells within total T cells, and (**E**) CD8α^+^ γδTCR^+^ T cells within γδTCR^+^ T cells were analyzed. Asterisks indicate statistically significant differences (* *p* < 0.05, ** *p* < 0.01; Dunn’s test).

**Table 1 viruses-17-00907-t001:** Comparison of the amino acid sequences of Meq in various Marek’s disease virus strains.

Strain	Accession No.	Virulence	Country	Position		
				71	77	326/385
Md5	AY243438	vv	US	A	K	T
RB-1B	AY243332	vv	US	A	K	T
BC-1 ^a^	AY362707	v	US	S	A	T
Jm ^a^	AY243331	v	US	S	A	T
CVI988 ^a^	AY243333-8	m	The Netherlands	S	E	I

Adopted from Shamblin et al., 2004 [43]. m, mild MDV; v, virulent MDV; vv, very virulent MDV; ^a^ Strains contain a 59-amino-acid insertion sequence in the transactivation domain.

**Table 2 viruses-17-00907-t002:** Primers used to introduce mutations into the *meq* gene.

Position in Meq	Substitution	Type	Sequence
71	Serine-to-alanine	Forward	5′-GAATCGTGACGCCGCTCGGAGAAGACG-3′
		Reverse	5′-CGTCTTCTCCGAGCGGCGTCACGATTC-3′
77	Glutamic-acid-to-lysine	Forward	5′-AGAAGACGCAGGGAGCAGACGTACT-3′
		Reverse	5′-AGTACGTCTGCTCCCTGCGTCTTCT-3′
326	Threonine-to-isoleucine	Forward	5′-AGGAAGCAGACGTACTATGTAGACA-3′
		Reverse	5′-TGTCTACATAGTACGTCTGCTTCCT-3′

**Table 3 viruses-17-00907-t003:** Primers used to construct the reporter plasmid.

Type	Type	Sequence	Reference
pp38	Forward	5′-GCTAGCGGCCGTGCCATTCTGAGAG-3′	[50]
	Reverse	5′-AAGCTTCCGTCCGACGAGAGCAAG-3′	
pp14	Forward	5′-TAAGAGCTCCTTATCCTATACCGCCGCCTC-3′	[50]
	Reverse	5′-AATAAGCTTGAGAGCATCGCGAAGAGAGAA-3′	
meq	Forward	5′-GCTAGCCCACGTACTGACGAATTTAGTAC-3′	[50]
	Reverse	5′-AAGCTTATTCTTAACATTCCAGCACCAAC-3′	
icp4	Forward	5′-TTCGAAGGGTTTAGGAGGGGCGCA -3′	[51]
	Reverse	5′-TGGTACCATTAGCCGCGACATCCATCT -3′	
gb	Forward	5′-TCAGATCTCAAGTCTCACTCACAAA-3′	[52]
	Reverse	5′-TCAGATCTGCTGTTCATAAATTGTGT-3′	
cd30	Forward	5′-TAAGAGCTCCTAATTAATAATAGCGTGCTC-3′	[53]
	Reverse	5′-TAACTCGAGTCCTGATCTCCCAGCATTGCA-3′	
bcl-2	Forward	5′-GCTAGCGACAGCCAGGAGGAAGCG-3′	[52]
	Reverse	5′-AAGCTTTGGGAGGGGGAGAGGAAG-3′	
il-2	Forward	5′-GAGCTCCGTCTTTGCAAACGATGACAG-3′	[54]
	Reverse	5′-AAGCTTAAATACAGCCAAAGATCAGTACT-3′	

Underlines indicate the restricted enzyme sites.

**Table 4 viruses-17-00907-t004:** Primers used for quantitative polymerase chain reaction.

Primer	Purpose	Type	Sequence
*icp4*	To qualify viral load	Forward	5′-GCATCGACAAGCACTTACGG-3′
		Reverse	5′-CGAGAGCGTCGTATTGTTTGG-3′
*i-nos*	Endogenous control for viral load	Forward	5′-GAGTGGTTTAAGGAGTTGGATCTGA-3′
		Reverse	5′-TTCCAGACCTCCCACCTCAA-3′
*meq*	To quantify transcripts of *meq*	Forward	5′-GTCCCCCCTCGATCTTTCTC-3′
		Reverse	5′-CGTCTGCTTCCTGCGTCTTC-3′
*β-actin*	Endogenous control for *meq* expression	Forward	5′-GAGAAATTGTGCGTGACATCA-3′
		Reverse	5′-CCTGAACCTCTCATTGCCA-3′
IRL inner	To check the restoration of the IRL region	Forward	5′-AGCTACCCCTTTCGGTTTGT-3′
		Reverse	5′-CACCCCCTTGTGGAAGTAGA-3′
IRL outer	To check the restoration of the IRL region	Forward	5′-CGAACGGAATGTACAACAGCTTGC-3′
		Reverse	5′-GATAAGACACTTTCCCACTCATAC-3′

**Table 5 viruses-17-00907-t005:** Summary of the results of the histopathological examination of tissues from chickens infected with recombinant Marek’s disease virus.

Virus	Sample Number	Lesions	Spleen	Bursa of Fabricius	Thymus	Remarks
rRB-1B_ Meq	#1	Tumor cell infiltration	++	+	++	Solid tumors were developed in the kidneys and ovaries. The spleen was noticeably enlarged. Macrophage infiltration was observed in the red pulp of the spleen.
		Meq^+^ cells	++	+	++	
	#2	Tumor cell infiltration	++	+	+	Enlargement of the spleen was noted. Macrophage infiltration was observed in the red pulp of the spleen.
		Meq^+^ cells	+	+	++	
	#3	Tumor cell infiltration	++	+	+	Enlargement of the spleen was noted. Macrophage infiltration was observed in the red pulp of the spleen.
		Meq^+^ cells	+	+	+	
rRB-1B_ Meq71/77	#1	Tumor cell infiltration	−	−	−	-
		Meq^+^ cells	−	−	−	
	#2	Tumor cell infiltration	−	−	−	-
		Meq^+^ cells	−	−	−	
	#3	Tumor cell infiltration	−	−	−	-
		Meq^+^ cells	+	+	+	
	#4	Tumor cell infiltration	−	−	−	-
		Meq^+^ cells	−	−	−	
	#5	Tumor cell infiltration	−	−	−	-
		Meq^+^ cells	+	+	+	

These results indicate the percentage of infiltrated tumor cells/Meq-positive cells per five fields of view. –, no significant change; +, <30%; ++, 30–80%.

## Data Availability

The datasets used and analyzed during the current study are available from the corresponding author on reasonable request.

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
