# Peer review of "Amino Acid Polymorphisms in the Basic Region of Meq of Vaccine Strain CVI988 Drastically Diminish the Virulence of Marek’s Disease Virus"

_viruses, 2025, doi:10.3390/v17070907_

Round 1
Reviewer 1 Report
Comments and Suggestions for Authors
The authors investigated how the known polymorphisms in the Marek’s disease virus (MDV) oncoprotein, Meq, impact its transcriptional regulatory functions, virus replication, and viral virulence in this study. They individually introduced three amino acid substitutions found in the MDV vaccine strain CVI988 or combined all three into a virulent virus-derived Meq and revealed that two polymorphisms in the basic region of the Meq protein significantly affect transcriptional regulatory functions and virulence. The experiments were well-designed and conducted, and the findings provide significant new additional insights on previous findings. However, some information is missing to make their experiments reproducible. Also, the text contained questionable descriptions and was unnecessarily redundant in multiple places.
Specific points:
Lines 42-59: Does this introductory paragraph effectively provide the necessary information for this study? If the author wants to explain the importance of cell population analyses in the infected birds, an introduction that focuses more directly on MDV infection would be appropriate.
Lines 131-132: A precise definition of the promoter regions used in the study is necessary for the reproduction of the experiments.
Line 228: The manufacturer of isoflurane is already stated earlier.
Lines 259-260: The fixation process is already described at the end of the previous section.
Lines 263-265: Were the slides counterstained with hematoxylin for hematoxylin-eosin staining?
Lines 277-279: Redundant to lines 228-231.
Lines 313-323: This entire section is almost completely redundant to the previous section and can be combined with it.
Fig. 1: It would be informative if the authors could provide the absolute luciferase activity (measured units) for the pCI-neo baseline either in the figure legends or the text.
Lines 382-384: Redundant to the Methods section.
Lines 422 and 628: The following observation contrasts with the previous one, but I do not think it is conversely related.
Fig 3: Are panels B and D necessary?
Figure 4: p-values in panel C must be explained in the legends.
Line 461: bursa -> bursa of Fabricius
Line 490: Although the authors state, ”These findings suggest…lower replication in vivo…", aren’t these more likely due to the lack of proliferation of latently infected transformed cells in the Meq71/77 infected birds?
Figs 6 and 7 and respective descriptions in the text: What do the absolute numbers shown in these figures represent? Do they indicate the number of respective cells in the entire spleen or thymus? Since the authors stained only 5X105 cells per well for FACS, they must have used numerous wells for each animal to obtain more than 107 positive cells. Was this the case? Additionally, DNA seems to have been extracted from the same organs for qPCR, which prevents the analysis of all cells from each organ. Please clarify.
Lines 567-569: Did the rRB-1B_Meq71/77 transform T cells though they did not increase? Also, did it cause a disease?
Line 576: Was the significance observed with rRB-1B_Meq71/77? Fig 7G looks to me to show the significance between the control and rRB-1B_Meq, not Meq71/77.
Fig 8E: The title for the Y-axis is incorrect.
Lines 715-718: The logic in this sentence is weak. What the authors demonstrated was the weak transactivation activity, and beyond that is mere speculation. The sentence structure does not reflect this logic.
Line 740: Although gamma-delta T cells are fewer in the thymus in the infected birds, this is not necessarily the case in the spleen and PBMC. Is the speculation about the susceptibility of gamma-delta T cells to cytolytic infection consistent with such observations?
Reviewer 2 Report
Comments and Suggestions for Authors
In this manuscript, the authors Jumpei et al. have investigated the amino acid polymorphisms in the basic region of Meq of CVI988 vaccine strain, and have demonstrated convincible evidences to show that the unique amino acid polymorphisms in oncogenic Meq protein, particularly at positions 71 and 77 differential from vvMDV strain RB-1B, contributed to the reduced virulence of vaccine. The experimental data is enough and solid. The MS is well written but several major revisions should be made before the acceptance for the publication in Viruses.
Major
1. For the Introduction, there are many citations are published nearly 15-20 years ago. It may be too old for the readers, especially for the first paragraph on MDV epidemiology. Some of the most recent and important studies on viral genome evolution, point mutations and virulence (Science. 2023 Dec 15;382(6676):1276-1281), and the emergence of variants of hypervirulent MDV (HV-MDV) in recent years (Viruses. 2023 Jun 25;15(7):1434; Viruses. 2022 Jul 27;14(8):1651) should not be ignored.
2. Lines 225, 249 & 414-420, usually as we known, a dose of 500 to 1000 PFU/bird of MDVs is enough for the virus-challenge experiments to induce MD and tumors, and the latter is a better choice for most researchers. But in this work, the authors used 5000 or 10000 PFU/bird for two bathes of animal experiments. It is needed to be explained and discussed.
3. Lines 386-388: the authors stated that the restoration of the IRL region in recombinant viruses have reconstituted from each BAC plasmid (rRB-1B_Meq and rRB-1B_Meq71/77) and was confirmed by PCR, but the data should be displayed rather than “data not shown”. In Fig.2A, Fig. S3 (RFLP) & Fig. S4A, designations of rRB-1B_Meq or rRB-1B_Meq71/77 are not in accordant to those described in the context. It is confusing whether the deleted IRL regions are restored or not?
4. Lines 467-478, for histopathological analysis, the stained sections and figures of immune organs in virus-challenged birds should be displayed as formal Figures (rather than a supplementary Fig.4A) in the context, together with Table 4. It is meaningful data for the readers.
Minor
In Fig 2B, rRB-1B_Meq71-77 should be substituted by rRB-1B_Meq71/77.
Reviewer 3 Report
Comments and Suggestions for Authors
The meq gene of Marek’s disease virus (MDV) encodes the Meq oncoprotein, which is responsible for the oncogenic properties of the virus in gallinaceous birds. Sequence differences in Meq correlate with differences in virulence of the virus. The attenuated MDV strain CVI988 (the gold standard vaccine against Marek’s disease) is non-oncogenic and non-virulent. Previous studies by the authors’ laboratory and other research groups, using recombinant viruses expressing chimeric Meq proteins, have shown that Meq of CVI988 is responsible for its attenuated properties. In the current study, the authors aimed to pinpoint the nucleotides involved in attenuation, by focusing on three unique polymorphic amino acids in the basic region of Meq, which differ between virulent MDVs and CVI988.
They introduced these three CVI988 Meq polymorphisms into the Meq of a very virulent MDV strain to investigate the effect on the transactivation and transrepression activity of Meq using in vitro reporter assays, the effect on virus replication in vitro, and the effect on viral replication, pathogenesis, and tumour development in vivo.
This is a very comprehensive set of experiments, which have been conducted thoroughly and carefully, with use of appropriate control groups, and the data have been analysed using appropriate statistical tests. The results provided strong evidence that polymorphisms at amino acids 71 and 77 of CVI988 Meq are sufficient to abolish MDV virulence, which is a significant finding for advancing the understanding of the role of Meq in MDV virulence and, as the authors suggest, might enable the development of more a more effective MD vaccine.
The paper is written very clearly, with good attention to detail and good use of the English language. It is organised well and is easy to read and clearly explains the purpose of each experiment. The Abstract provides a good overview, and the Introduction gives a thorough background to the study and makes the aims clear. The Materials and Methods are detailed and clear, the Results are presented clearly in good tables and figures. The Discussion is detailed and interprets the results in the context of the relevant literature, speculating on how these polymorphisms may affect virulence and suggesting future work to investigate this.
I have no criticisms, and just a list of minor queries and suggestions to improve clarity:
- Introduction: it would be useful to include the ICTV classification of MDV (subfamily Alphaherpesvirinae, genus Mardivirus, species Gallid alphaherpesvirus 2).
- Introduction: It would be useful to summarise the MDV pathotypes (m, v, vv, vv+) here.
- Introduction line 78: make it clear here that CVI988 is an attenuated MDV.
- Introduction: line 78 refers to the Threonine/Isoleucine polymorphism as ‘position 326’, but in line 79 it is referred to as ‘position 385/386’. It needs to be made clearer that these refer to the equivalent positions in virulent MDVs and in CVI988 because of the insertion in CVI988.
- Meq is a double-copy gene, present in both the IRL and TRL repeat regions of the MDV genome, but this has not been mentioned. In generating the recombinant RB-1B MDV, were the CVI988 polymorphisms introduced into both copies of the meq gene of RB-1B or just into the TRL copy? If the latter, the authors should discuss how this affects interpretation of the in vivo results.
- Three polymorphisms (positions 71, 77, and 326) were investigated in the reporter assays, but only the position 71 and 77 polymorphisms were introduced into the recombinant virus. Why was the 326 polymorphism not included? Probably because the reporter assays showed it was less important than 71 and 77, but this should be made clear.
- Table 3: it would be useful to have an additional column to state the purpose of the four sets of primers (ICP4 to qualify viral load; i-nos as endogenous control for viral load; meq to quantify transcripts). What was the purpose of the B-actin primers? B-actin qPCR is not mentioned in the methods or results.
- Why was 5000 pfu MDV used in experiment 1 and 10,000 pfu in experiment 2? Possibly to check whether rRB-1B_Meq71/77 showed any pathogenicity when used a high dose – this needs to be made clear.
- Data for lymphoid organ weights are presented in the results, but there is no mention in the Materials & Methods of recording body weights or organ weights. This could be added to section 2.8.1.1.
- Figure 2B shows viral load expressed as viral genome copies per million cells. However, the Materials & Methods (line 284) states, ‘Viral load values are presented as the ratio of icp4 to i-nos’.
- Legend for Figure 2C states, ‘mRNA expression of each Meq isoform in CEFs infected with rRB-1B_Meq and rRB-1B_Meq71/77 was confirmed by reverse transcription-qPCR’. But there is no detail about this RT-qPCR in Materials and Methods.
- It would be very interesting to test whether rRB-1B_Meq71/77 is an effective vaccine against MD in an in vivo trial, and to compare its efficacy with CVI988.
- As another point of interest, Fiddaman et al, 2023 (Science, 382:1276-1281, Ancient chicken remains reveal the origins of virulence in Marek's disease virus) identified the 71S to A polymorphism as being one of the key differences between the ancient MDV strain and Md5/RB-1B, and the transactivation ability of the ancient strain was clearly lower, supporting the results in Sato et al’s manuscript, while also demonstrating that the S allele is the ancestral allele. The authors may wish to include this in their Discussion.
Round 2
Reviewer 2 Report
Comments and Suggestions for Authors
The authors have addressed reviewer's concerns and is acceptable for the publication.